# Ethical Decision Making in Iot Data Driven Research: A Case Study of a Large-Scale Pilot

**DOI:** 10.3390/healthcare10050957

**Published:** 2022-05-23

**Authors:** Sofia Segkouli, Giuseppe Fico, Cecilia Vera-Muñoz, Mario Lecumberri, Antonis Voulgaridis, Andreas Triantafyllidis, Pilar Sala, Stefano Nunziata, Nadia Campanini, Enrico Montanari, Suzanne Morton, Alexandre Duclos, Francesca Cocchi, Mario Diaz Nava, Trinidad de Lorenzo, Eleni Chalkia, Matina Loukea, Juan Bautista Montalvá Colomer, George E. Dafoulas, Sergio Guillén, María Teresa Arredondo Waldmeyer, Konstantinos Votis

**Affiliations:** 1Centre for Research and Technology Hellas, Information Technologies Institute, 57001 Thessaloniki, Greece; antonismv@iti.gr (A.V.); atriand@iti.gr (A.T.); kvotis@iti.gr (K.V.); 2Life Supporting Technologies, E.T.S.I. Telecomunicación, Universidad Politécnica de Madrid, 28040 Madrid, Spain; gfico@lst.tfo.upm.es (G.F.); cvera@lst.tfo.upm.es (C.V.-M.); jmontalva@lst.tfo.upm.es (J.B.M.C.); mta@lst.tfo.upm.es (M.T.A.W.); 3Iniciativa Social Integral SLU, 46002 Valencia, Spain; jmlecumberri@isibenestar.com; 4Mysphera SL, 46980 Paterna, Spain or psala@mysphera.com (P.S.); sguillen@mysphera.com (S.G.); 5ITACA Institute, Universitat Politècnica València, 46022 Valencia, Spain; 6Lepida Scpa, 40128 Bologna, Italy; stefano.nunziata@cup2000.it; 7Azienda Unita’ Sanitaria Locale Di Parma, 43125 Parma, Italy; ncampanini@ausl.pr.it (N.C.); montanarienriconeuro@gmail.com (E.M.); sifrana@yahoo.it (F.C.); 8Leeds City Council, Leeds LS2 8BB, UK; suzannemortonemail@gmail.com; 9Centre Expert en Technologies et Services pour le Maintien en Autonomie a Domicile des Personnes Agees, 75015 Paris, France; alexandre.duclos@madopa.fr; 10ST Microelectronics, 38000 Grenoble, France; mario.diaznava@st.com; 11Cruz Roja Española, 15002 A Coruña, Spain; delorenzo@cruzroja.es; 12Centre for Research and Technology Hellas, Hellenic Institute of Transport, 57001 Thessaloniki, Greece; hchalkia@certh.gr (E.C.); mloukea@certh.gr (M.L.); 13Digital Cities of Central Greece-CitiesNet, 42100 Trikala, Greece; gdafoulas@med.uth.gr

**Keywords:** IoT, Active and Healthy Ageing, participatory, deliberative, ethical and sustainable, decision making, older adults

## Abstract

IoT technologies generate intelligence and connectivity and develop knowledge to be used in the decision-making process. However, research that uses big data through global interconnected infrastructures, such as the ‘Internet of Things’ (IoT) for Active and Healthy Ageing (AHA), is fraught with several ethical concerns. A large-scale application of IoT operating in diverse piloting contexts and case studies needs to be orchestrated by a robust framework to guide ethical and sustainable decision making in respect to data management of AHA and IoT based solutions. The main objective of the current article is to present the successful completion of a collaborative multiscale research work, which addressed the complicated exercise of ethical decision making in IoT smart ecosystems for older adults. Our results reveal that among the strong enablers of the proposed ethical decision support model were the participatory and deliberative procedures complemented by a set of regulatory and non-regulatory tools to operationalize core ethical values such as transparency, trust, and fairness in real care settings for older adults and their caregivers.

## 1. Introduction

The massive revolution of the Internet of Things (IoT) holds the promise of using advanced devices, sensors, and communication platforms to share data and create novel services and applications towards improving the quality of life [1] and addressing the health needs of the ageing population [2,3]. In recent years, several relevant research works exploited the power of various emerging wireless technologies that measure health parameters in order to provide pervasive health monitoring services and to facilitate remote medical management [4,5].

Nonetheless, the use of open and easily accessible IoT ecosystems raises ethical, security, and privacy challenges, particularly for the healthcare domain [6,7,8]. The ethical concerns of user engagement with IoT technologies, specifically those health related (H-IoT), are associated with inherent risks of the IoT devices, including the importance and accuracy critical to the delivery of health data. In this sense, a more systematic assessment of users’ opinions and beliefs is required from researchers in order to (a) gain insights about a proactive engagement in ethical issues within an IoT landscape and (b) provide accurate and timely awareness to all stakeholders to support the comprehensive and smooth processes of ethical decision making and assessment [9].

Considering the scale of interconnection through IoT technologies, great importance is placed on the moral implications concerning data management of IoT systems. Relevant literature presents the important ethical challenges that can emerge from IoT technologies data and their interactions [10]. More specifically, in the big data era powerful analytics tools allow the collection and analysis of large amounts of data to identify patterns in the behavior of groups and communities. This can also affect decisions that concern the interrelations of communities, hence the individuals involved [11]. The ethical design of IoT and H-IoT services has been based on unobtrusive monitoring applications and considered as a crucial process to protect user autonomy according to their sense of personal identity and decision making. Moreover, even in the case of sophisticated algorithms that inform decisions, individual aspects, which should not be subject to the risk of profiling, are legally provisioned. However, IoT service providers, as well as consumers, do not have a clear picture of the available legal provisions [12].

During recent years, an increasingly growing field of research and practice has been focused on the analysis of ethical perspectives in the use of IoT technologies including ethical theoretical frameworks and conceptual models to identify the ethical challenges coming from the implementation of IoT initiatives [13,14,15]. However, few of these research works have addressed the ethical processes ad hoc enmeshed in technological IoT innovation and related to the production and management of enormous amounts of data related to human behavior characteristics. The real ethical implication lies in the fact that this process of IoT data manipulation is conducted automatically and often in the absence of any human interference [16]. Moreover, the high number and heterogeneity of devices and technologies in IoT based environments can affect the decision-making process, either by offering support or becoming excessive, and can lead to an over-controlled environment [17].

In accordance with these concerns the notion of data quality must be highly considered in the case of the collective approach of studying and predicting group behavior, rather than in profiling single users, where there are also concerns of invasion and discrimination in indirect data-driven decisions. The relevant literature [18] suggests that data transparency is closely related to data quality and enhances the decision making and planning, while acknowledging the need for a comprehensive framework of data transparency.

Also, according to Diène et al. [19], certain decisions need to be made on aspects such as how frequently data should be captured, how many data are captured, and their need to be archived. However, there is still a lack of description of the intermediate stage, where decisions delineated in ethical frameworks, made either by humans or machine intelligence, are ‘translated’ from theoretical models into daily research practice, supporting the process of ethical decision making. Trust and transparency in personal data management are critical components of the decision-making process.

Above all, specific consideration and thinking around the ethical issues is required considering the values that need to be promoted regarding the use, sharing, and re-use of big data in a research context [20]. In this respect, the importance of having ethical frameworks adapted to the context of handling big and sensitive data has been acknowledged and is crucial [21]. However, although a few ethics frameworks have been introduced so far for working with big data in health research, the ethical decision-making process has not been sufficiently analyzed yet.

If good decision making is a complex endeavor, ethical decision making particularly for health contexts is more demanding [22,23]. Therefore, decision support models for diffused responsibilities in large-scale research is needed. Nonetheless, a model by itself is not sufficient if feedback options of data management are not communicated between the diverse stakeholders of the research (i.e., researchers, developers, clinicians, end users) to reflect in an understandable and instructive way participatory and deliberative ethical guidelines.

The more recent literature advocates the importance of participatory assessment of technologies which aim to advance the role of public and users for decision making considering ethical dimensions in diverse contexts [24]. However, in the context of LSP ecosystems, the implementation of overarching principles in practice as envisioned not only by legal provisions but also guided by ethical requirements is well recognized in the context or Responsible Innovation (RI) [25].

The research work described in this manuscript informs on a novel ethical decision-making framework for large-scale pilot managers, particularly in the Active and Healthy Ageing (AHA) IoT based solutions domain. This framework has been defined by the ethical board in collaboration with law consultants within ACTIVAGE-, a H2020 large-scale pilot project [26]. ACTIVAGE created the first European IoT ecosystem, enabling the deployment and operation at a large scale (i.e., in nine deployment sites (Table 1) complemented by new pilot sites in the context of open calls and IoT technology exploitation. Overall, the DSs involved more than 7000 people around Europe, Spain, France, Italy, Germany, Greece, Finland, and the United Kingdom) of Active and Healthy Ageing (AHA) IoT based solutions and services to respond to real needs of caregivers, service providers, and public authorities.

The proposed ethical framework has been the first initiative to address the ethical aspects of data management and governance in the context of AHA and IoT based solutions. The problem-driven decision-making process was outlined and highlighted the real obstacles emerging along the way in the context of a highly diverse and multidisciplinary scientific team.

## 2. Materials and Methods

The proposed ethical frame of decision making was developed through a bottom-up approach, anchored in the analysis of a large number of stakeholders. Approximately 50 different partners of academia, universities, enterprises, (big companies as well as SME businesses), administration (city councils, regional authorities), social enterprises coordinating end users, as well as European citizens comprised the large-scale pilot.

A co-creation framework [27] was introduced to assess the needs, preferences, and perceptions regarding user acceptance, trust, confidentiality, privacy, data protection, and safety. In this context, the management of information ethics by telecommunication engineers, sociologists, health technicians, social workers, caregivers’ data protection officers, and many more, posed significant challenges for deliberate exercises in ethics and decision making due to the heterogeneity of data and diverse disciplines, cultures, organizations, and social networks. From the early beginning of the project, it is considered of critical importance that beyond the adherence in the letter of law, it is essential to review the ethical dimension of data processing by initiating and elaborating further an ethical framework based on several ethical values and principles. A data ethics framework has been defined to map out the ethical principles for data proper use, to maximize the value of data, and to maintain natural persons’ rights towards a sustainable IoT environment (Figure 1).

Specific attention had also to be paid to the analysis of information along with the implementation of core ethical principles such as transparency and accountability and trust and confidentiality, which determined the decision–making process in a complex socio-technical context.

Understanding and operationalizing the ethical decision-making process initially brought the concern to the research communities to become increasingly aware of the ethical and social perspectives and implications that IoT research in overall and each technological component may bring. To this end inner and outer feedback loops have been activated to build and use iterative and sustainable sources for this awareness.

Outer feedback loops were based on vertical and horizontal experiences and know-how from synergies with other LSPs, as useful insights in the project IoT community. Inner feedback loops were anchored on training and education through practical tools as online courses to achieve data management quality. The decision-making process was based upon the methodology of feedback loops (Figure 2) between the various phases of the project and between the diversity of implications that designers and developers face at the local level. However, harmonized action points of internal and external mechanisms are based on deliberative ‘dialogue’ between the sociotechnical groups.

Overall, the diverse piloting environments were treated as ‘living organisms’ that had to go beyond the formulaic approaches of ethical norms and principles in implementation and were compelled to apply practical assessment and monitoring tools in a consistent way to achieve ethical awareness of piloting activities operation. In practice, the specific context of IoT technologies and the related constraints had to be considered, whilst corresponding action plans about ethics management supported this process and were required to be applied in consistency with the initial planning from the early beginning of the project until its finalization. Nonetheless, the main determinants of our ethical design were twofold: (a) to consider the requirements of project pilots or deployment site performance across several European countries both at an individual and at a project level, striving to find convergence points; and (b) given that technological artefacts were not neutral, intermediaries had to be designed and used upon a moral decision making as a joint effort of individuals.

The proposed model of ethical decision making was composed of 4 stages: (a) develop the frame; (b) assess through deliberative dialogue; (c) apply and update during the whole lifecycle of pilot activities; and (d) evaluate.

In all phases, a placeholder was relevant knowledge in clear decision- making processes to the sensitive target group of older adults. This action was enabled by previous initiative s, such as the Alliance for Internet of Things Innovation (AIOTI) Working Group 5 [28], which focused on Smart Living Environment for Ageing Well and had been equally acknowledged by all sites.

Also, effective enablers such as social responsibility and solidarity have been recently activated towards handling the risks of uncertainty and anxiety for seniors and communities caused by the unprecedented world health crisis due to COVID-19. The main contingency measures that the confinement of COVID-19 forced all deployment sites to adjust their pilot activities were reported in ethical reports indicating how this new situation of COVID-19 had been handled ethically.

End users provided their feedback to the LSP in many ways by testing the IoT solution, giving feedback on improvements and software solutions, and providing their views on social and ethical aspects of IoT apps and services’ use.

In this line, apart from the main contingency measures, after a careful analysis of user needs and requirements, some pilot sites developed additional platforms, such as the AgeWare platform of Deployment Site (DS) of Sofia, to fit the current needs of the deployment site. This caused the development of additional functionality and services to exceed the initially planned use case services. Full UI/UX for web, tablet and smartphone, online questionnaires, daily schedule, interactive TV, videoconferences, fall protection, and additional health wearables were deployed; therefore, the need emerged for compliance with legal requirements with the support of a robust infrastructure (block chain-based) in order to make the respective platforms operational.

To harmonize the action plans of diverse contexts in respect to ethics, sustainable ethical models were created to allow common understanding of data processing chains, their value, and proactive measures. These models were initially the outcome of the experience gained from brainstorming events and collaboration on scientific publications (i.e., White Paper, ‘Personal Data Protection for Internet of Things Deployments: Lessons Learned From the European Large-Scale Pilots of Internet of Things’) in the context of European Commission research programs emphasized on the Internet of Things (IoT) and data protection in large-scale pilots (LSPs). In the context of IoT policy frameworks for trusted, safe, and legal environment for IoT and in collaboration with the activities groups (i.e., Activity Group 5).

As a follow-up activity, two experts, one with expertise in system-level design methodologies and one in ethics, took over the role to coordinate data protection at the project level (Mezzanine Model). Thus, they set up a series of exploratory bilateral meetings with pilot site leaders and persons responsible for ethics in each pilot. The main aim of the activity was to investigate IoT risks and gain insights in sharing and protecting information collected and stored in IoT platforms.

Knowledge from diverse LSP methods and techniques for managing critical ethics issues was extracted. These issues included data subject rights against business and organizations and issues related to data business models. Methods and techniques for managing these issues in the LSPs were employed to address the specific challenges of ‘smart living environments for ageing’ use cases. This exploitable knowledge asset was enriched by security and privacy topics incorporated in a technical workshop held. Both scientific and technical members had to leverage on the project knowledge aggregated about risk assessment and confrontation along with the progress of ethics and legal activities at the project level. The main results of these brainstorming sessions of security and privacy were used as the base for the design of specific instruments such as security, privacy, and ethics constructive surveys.

Essentially, these surveys were in liaison with data analysis rules such as need for DPIA interpretation beyond legal requirements and GDPR and clustering risks and impacts to generate data models and formulate common rules of data privacy and protection.

These instruments for assessing issues related to privacy and security have been designed and constructed by a multidisciplinary team consisting of professionals with well-grounded experience in data analysis and mixed methods (quantitative data and qualitative data, external ethics, and legal experts). More specifically, a concrete proportion of data analysts and ethics experts has been defined according to pilot scale activities to be dedicated to the development, testing, and use of IoT technologies along with the sensitivity of data that must be handled. Therefore, it has been considered of high importance to assign one pilot responsible per site and one ethical manager, approximately 30% of the members involved in pilot activities to provide IoT-based solutions of perceived value for smart living of older adults. It is worth mentioning that the core ethic members of the project are involved in European initiatives [29]. The process of research activities/benchmarking/data analysis was carried out by 70% of the partners involved in the research activities.

This work has also been complemented by the thorough study of the relevant literature concerning ethics and security questionnaires to provide better insight into diverse ethical concerns in IoT ecosystems and landscapes.

Therefore, the specific instruments comprised a set of three tools: (a) Questionnaire for the identification of the main ethical issues per topic to be considered in the pilot sites; (b) a Data Protection Impact Assessment (DPIA) evaluation; and (c) the identification of a best and worst practices survey. Further details about each instrument are provided below:

### 2.1. Identification of Main Ethical Aspects

The first tool developed for the framework was aimed at identifying the specific ethical aspects to be considered in the pilot sites. For this, the pilot sites were asked to fill in a table that contained a list of topics to assess, together with some fields for proposing a description of the item, its status, and a possible mitigation strategy.

The list of elements comprised a total number of nine items, seven of them being generic ones and two of them specifically related to the ACTIVAGE project (i.e., Items 8 and 9): (1) ethics approval processes, data sharing agreements and NDAs (non-disclosure agreements); (2) data protection processes; (3) health and safety regulations and standards; (4) ethics policies in each organization; (5) recruitment issues; (6) legislative issues; (7) participants and informed consent; (8) expand phase: co-creation (use cases’ exchange) and ethical and legal implications; (9) growth phase: open calls and data protection principles (i.e., safeguards in case of data re-use).

To facilitate the completion of the table, pilot sites were provided with a questionnaire for analyzing each of the specific topics (see Appendix A).

### 2.2. Data Protection Impact Assessment (DPIA)

To increase the societal and legal acceptance of an IoT system, it was essential to guarantee that the deployed system would fully respect users’ right to privacy and would act in accordance with the local and international laws on data protection and end user privacy. The followed approach was to conduct a security risk analysis using the STRIDE (spoofing, tampering, repudiation, information disclosure, denial of service, elevation of privilege) approach, a security threat analysis approach at every level of the system architecture and its deployment approach: (a) device, (b) gateway, and (c) cloud application domain.

Also, the privacy methodology was defined using the concept of DPIA (Data Protection Impact Assessment) as a key methodology introduced in the General Data Protection Regulation towards the management of fundamental human rights during the processing of personal data.

The DPIA evaluation was not administered solely as a mandatory legal tool for data privacy risk assessment. Its interpretation was used to reflect risk identification and the impact of data use, collection, and storage in different contexts.

In respect to security and privacy risk mitigation measures, a main concern was the initiation and reservation of safe databases. For this reason, IoT secured devices and gateways as well as secured platforms with controlled access (e.g., through eIDA certificates for professionals and login/password use of one-time pin codes for users/patients and carers) were utilized. Documented processes and procedures along with data sharing agreements were an integral part of the efficient handling of data protection issues as preventive mechanisms at an early stage and in case of data breach incidents.

### 2.3. Metrics and Checklists: Best and Worst Practices

The third tool included in the framework was a questionnaire to assess the best and worst practices in the IoT environment over five critical topics: (1) organizational measures and user recruitment; (2) data processing security; (3) large-scale projects; (4) ethics; and (5) business areas. The questionnaire assessed pilot managers’ opinions, attitudes, and perceptions and had both a quantitative (Likert scale based) and qualitative (descriptive) part.

The main objective of this survey was to gather end users’ opinions specifically on ethical issues according to the context of (a) IoT technology use in general and (b) IoT devices/platforms. Therefore, an ethical impact assessment survey was designed with concrete areas of concern, in order to exploit the empirical evidence of persons/entities involved in the process of ethics evaluation and to support the process of ethical decision making.

### 2.4. Evaluation Process

In this line, awareness about the use cases and data related to each specific use case of the study has also been provided since according to some of them, the users of technologies that belonged to the experimental group had to authorize the sending of data to the different applications of the pilots. This authorization was explicit, and the user had to accept the policy guidelines of data usage towards participation acceptance.

The selection and recruitment of users was considered a crucial part of the user involvement process and was expected to affect the quality of the outcomes and the sustainability of the research. Therefore, a satisfactory number and combination of user characteristics was sought in the frame of gender balance and equality. Moreover, participant familiarity with IoT services and platforms was among the high considerations in the recruitment design depicted on the methods of their engagement (i.e., use of consent forms through infographics). Additionally, health and clinical data use was explicitly documented in alignment with users’ rights related to data portability, power, and freedom to be exercised and respected.

Older adults’ control over their personal data was among the main concerns of IoT service providers and researchers. The ownership of data was among the critical questions with ethical implications about the freedom and dignity of users. This was valid before, during, and after the research, as well as how to consider users’ data control beyond the original datasets, namely data that was produced after the initial recordings.

## 3. Results

A process of shared visions in respect to ethical and societal considerations of IoT ecosystems was carried out through the project duration, a period of 46 months, to provide AHA services and solutions to 7000 users including older people, their caregivers, and key actors from the healthcare domain. To ratify the deliberative process of decision making, social and technical actors proceeded to the design and construction of instruments upon two main clusters: (a) the regulatory and (b) the non-regulatory items.

The main users of these surveys were the pilot managers themselves, together with the rest of the stakeholders involved in the deployment and management of each pilot. After this assessment mitigation and correction, actions had to be suggested and followed by the pilot-responsible people. Therefore, the main beneficiaries were the end users (older adults and their caregivers).

The items of the existing questionnaires were selected in respect to the significance of (a) ethical, (b) security, and (c) privacy concerns and their complexities mainly reflected in the decision-making practice.

Therefore, three areas have been investigated: (a) the identification of the main ethical issues per topic; (b) regulatory instruments such as the DPIA evaluation; and (c) the identification of best and worst practices.

The questionnaires provided for administration to all local pilot sites are described below.

### 3.1. Ethical Issues Per Topic

Throughout the whole duration of the pilots, all pilot sites monitored and identified different ethical issues together with status, plans, and mitigation strategy. The figure below (Figure 3) summarizes the aspects identified per topic. As it is evident from Figure 2, ‘data protection processes’ and ‘informed consent’ issues are considered as the most critical ones, followed by ‘ethics approval processes’. These were expected since they are in line with the GDPR key policies. The emphasis on recruitment issues, however, has been unexpected and may also hide equity and gender issues behind it.

### 3.2. Privacy and Security Issues

The evaluation of the data management activities across the project pilots provided the following results in terms of data processing and privacy and security risks and threats.

According to Figure 4, none of the privacy risks are seen as very critical by most of the project pilot sites. As critical are considered the number of third parties that have access to the data as well as the data processing, which are indeed valid areas of concern and require in the future further privacy and policies to be established for them.

Regarding security risks, only mobile interfaces and cloud interface security issues were found very critical and only from one pilot site (Figure 5). Additionally, critical security risks regarding network services and insufficient authentication were considered as critical only from one pilot site. On the contrary, most security issues were considered to be slightly or non-critical at all from the vast majority of the pilot sites. Web interface security is viewed as moderate criticality, but there are technologies (i.e., block chains) to cover such risks.

### 3.3. Best and Worst Regulatory and Non-Regulatory Practices

Evaluation of the data management activities across all pilots provided the following results in terms of three main topics: organizational measures and user recruitment, large-scale pilot management and ethical framework, and some final evaluation of the proposed ethical framework.

Most of the pilot sites found all organizational measures and user recruitment guidelines as effective or very effective. Among the most effective methods to finalize regulatory activities were: (a) volunteers/participants recruitment process, (b) consent procedure, and (c) roles identification and responsibilities allocation, which helped to prepare the ground for DPIA assessment (Figure 6). The consent procedure effectiveness though seems to be able to be improved.

### 3.4. LSP Concerns in Data Management

The evaluation of the ACTIVAGE LSP data management aspects was rated as acceptable to good/very good; thus, no major improvements were needed (Figure 7). The only part where improvement could be considered was the methods for data curation and analysis and the data management strategies and policies, as indicated by one site. In this case, it should be mentioned that additional explanations were requested, and the justification was the following: qualitative description of research data curation and use practices in institutional repositories.

The data governance was also very positively evaluated and none of the aspects were considered to require improvements (Figure 8).

The same as the LSP data management aspects are also the LSP data government parts, which are again viewed as acceptable to good/excellent by most pilot sites. Only the data integration and interoperability issues were considered to need improvements by one pilot site, but the rest of the sites considered this aspect as good with no need of improvements.

Lastly, the measures for data management were rated as good and acceptable by the majority of the pilot sites without the need of implementing improvements and only one of them highlighted the need to improve the effectiveness of the proposed tools (Figure 9).

### 3.5. Final Evaluation of ACTIVAGE Ethical Framework

Finally, the whole global ethical framework was evaluated in terms of a series of values in nine of the pilot sites (Figure 10).

In general, the ACTIVAGE ethical framework was very well received with most experts considering that it may enforce the rights to private life, but without providing any supporting evidence on it. The obtained results showed very positive assessments of all the aspects evaluated and only one of the pilot sites rates negatively on two of the aspects (commercialization of de-identified information and enforcement of the right of private life), while all the others consider these aspects as following the UCM.

## 4. Discussion

So far, empirically developed strategies based on known models, such as the salutogenesis model [30], aiming to strengthen the thinking and capability of patient care and health have been widely used. Given that the idea of health occupied a central space as much as technologies, this model supported the procedure of ethics integration according to the assets of well-being and health monitoring. Nonetheless, concerning the healthcare domain and the solutions addressing older people software developers often fail to address user preferences, ultimately implementing non-user-friendly systems [31].

To address this inadequacy in the context of our study, attention was given to the ethical requirements and ethical dimensions of decision making when working with this vulnerable group. For instance, the necessity for identifying the proper design of such interfaces was one of the key factors. It is highly recommended for such technological solutions to involve the main social and technology actors along with the older adult users as actively as possible from the beginning of the design and development process. For this reason, co-creation has been used as an efficient design process and as an ethical approach incorporated to the user engagement process.

In the process of identifying ethical aspects for our framework, human values with ethical importance had to be considered in a more systematic way in the technological design. This was achieved through: (a) participatory design as a commitment to a democratization of the process and (b) human welfare, as well as value sensitive design by the developed technology, which accounts for human values [32].

The AHA pilots focused on the specific needs of older people such as the early prevention of mental, behavioral, and health-related decline, which was achieved by introducing seamless behavioral assessment and monitoring devices and intelligent algorithms both at home and in smart city environments. Notwithstanding, despite the unobtrusiveness, surveillance technologies could be ripe for conflict if not used in alignment with older adults’ needs and preferences [33]. In addition, the development of devices without the explicit consideration of ethical implications and ethical values that should be implemented could limit their adoption and efficacy.

The main objectives were to identify specific IoT ethical hints arising from the diverse use cases of the pilot sites, information management of raw data, and wider moral problems related to metadata, to assess their impact by receiving feedback from the key actors of the study, and to apply measures to combat the identified ethical risks. Specifically, health IoT data raises several ethical concerns for privacy and confidentiality stemming from IoT enabled devices, the ad-hoc sensitivity of health-related data, and their reflection in healthcare services’ delivery. Thus, three different attributes have been addressed: (a) ethics of datatype, (b) ethics of data concerning data (metadata), and (c) ethics of human interaction. Specifically, apart from the need to initiate mechanisms for regular and systematic data management in each pilot, data had to be addressed according to curation, processing, dissemination, and sharing regulations to initiate morally acceptable solutions (e.g., right conducts or right values). Additionally, big data associated with healthcare apps and devices in smart environments for vulnerable groups, such as older adults in the EU policy context, required further analysis beyond mere ethical challenges’ identification. Researchers and healthcare personnel had to adapt ethical principles and strategies such as discussing intellectual property and focus on articulation and practice of interoperable technologies by applying values and virtues rather than focusing on the consequences of the technology use ad hoc.

In the frame of the study and beyond, an ethical approach has been integrated into device design, pilot operation initiation and enrolment, actor involvement, and work organization. Co-creation was not only a safe measure to guarantee the end users engagement with innovative service development and implementation, but also an integral part of an efficient design process of the whole chain of technological solution implementation intended to be installed in older adults’ homes. Active user engagement from the outset was supported during all the pilot phases.

Participation of older adults, their caregivers, and professionals in the evaluation of every phase of AHA service was part of this co-creation process, as was close collaboration between all parties to create new evidence of the value of IoT-enabled AHA. From an ethics perspective, this meant aggregating and assessing local dimensions first, but then making them available and interoperable at a global level to make sure that solutions could be exchanged across regions without compromising the ethical dimension. Likewise, importing solutions from third parties or exporting solutions into new settings required having an ethical framework that could be adopted by new parties to comply with the ethical assessments and needs that had been gathered in the previous phases. Moreover, apart from the users’ engagement in pilots at the individual level, social engagement was encouraged via the use of social participatory apps, connecting older adults to smart home infrastructures to provide data about events and linking them to other peers. Overall, the demonstration actions related to user engagement were inclusive and representative of various user groups.

The main outcome of the ethics perspectives analysis in respect to large-scale pilot and planning was the identification of critical key areas with ethical implications and corresponding key ethical issues, risks, and data ethics evaluation during the different phases of the process. Ethical assessment was closely related to user-centered design which targeted older adults’ and their carers’ quality of life and care system efficiency gains.

An initial activity was to interpret the theoretical and conceptual values in respect to individual autonomy, dignity, and human rights in common actionable, credible, and trusted strategic mechanisms developed upon a ‘common ethos’ for data management and seniors’ engagement in a single common interoperable IoT ecosystem across a multi- stakeholder ecosystem. Therefore, researchers with the constant support of legal consultants and internal/external experts had to consider the meaningfulness of the provided technologies, the appropriateness of being used by similar target groups, and the purpose of using them. Consequently, ‘ethical thinking’ requires that concrete values had to be implemented during the usage of IoT technologies, apps, and services for improving seniors’ quality of life.

Despite the common ethics norms and goals, the diversity in scenarios and use cases as well as national law requirements brought a variety of challenges, and each pilot site had to display relevant flexibility, along with experience and abilities, to meet these challenges. Researchers and developers had to elaborate within their local pilot where ethical considerations were raised regarding the specific use cases and the entities of IoT ecosystems such as users, service providers, software/hardware providers, standardization organizations, policy makers, etc. However, several action points were common and mandatory in specific scenarios such as the detection of anomalies or of abnormal trends, the prevention of dangerous conditions, and the early detection of relevant issues and their ethical dimensions. Among the concerns was the question of assessing whether the detailed analysis of data tracking was sufficient to avoid ethical implications mainly according to health data monitoring. These considerations had to be traced to the general level of project considerations to maximize the value of technologies that would enable better communication between the user, the caregivers (both formal and informal), and the responsible researchers.

Informed decisions about big data and health data were an effort to balance the power and vulnerability of the sensitive data. The model of inclusive decision making followed upon the principles of accountability and experience reflection. Ethical awareness of the specific problem was the initial step in the ethical decision-making process. Decision makers of each deployment site, after a thorough internal study, communicated the group’s position and interests at the project level for further consideration to receive feedback from relevant ethical merits, to assess, to communicate the decision in a transparent manner to all partners, and finally to implement the strongest ethical opinion at the local level.

Considering the large pilot performance, the protection and trust of data along with fundamental issues of ethics in compliance with user needs and rights, especially in cross- border data management, had to be carefully tackled. Moreover, the intelligence, speed of transmission, and interconnectivity of IoT devices, apps, and systems, along with fragile group dependence on technology, brought the urgent need for clear guidelines that would guide, cultivate, and assess the ethical behavior of researchers to establish transparency and trust. Mainly moral problems were identified in relation to data generation, recording, curation, processing, dissemination, sharing, and using, as well as to specific clusters of information gathered from specific apps/devices.

While not denying the importance of security and privacy, the most challenging issue was the engagement with the notion of ethics in the context of highly diverse data, different reference use cases, data sources, devices/apps, services, and sub-service provision. In the landscape of IoT technology use, the multidisciplinary pilot sites had to cope with location data and data generated and collected from sensors and mobile devices, apps, and platforms. In addition, internet connected devices provided granular information of personal data uploaded in cloud storage and linked to social networks, something that could potentially allow the identification of user behavior through context information. In addition, ethical challenges emerged from IoT equipment installations at users’ homes.

In terms of making IoT research ethically acceptable, all the involved societal actors and innovators had to indicate mutual responsibility in the light of scientific innovation smooth implementation in the daily routines of older adults [34]. Despite the needs of a high-level security approach in the use of IoT, the risk of using huge amounts of data by introducing algorithms had to be reduced. Concerning the value of data about user behaviors collected from all solutions in the different deployment sites, social values had been embedded to align ethics with the use of the Internet of Things.

Finally, it is worth mentioning that the IoT ecosystem of the study was a network of stakeholders that worked together in deploying and testing innovative IoT solutions and collecting evidence to motivate refinements and decisions towards broader investment and adoption. A multidisciplinary team of technologists, social and healthcare professionals, enterprise managers, scientists, and entrepreneurs transcended local boundaries and scaled up to a European level and enabled an effective exchange of experiences and cooperation among peers (e.g., users, providers, policy makers) grouped into various clusters assigned to gather and handle data for smart assisting living [35] of seniors across Europe. This multidisciplinary character of the scientific community enabled the generation of evidence on remote monitoring as well as applications and services for supporting the independent life of older people at home.

## 5. Conclusions, Limitations, and Future Research

During recent years, the importance of ethics in European projects or initiatives has been growing in the same way that once happened with gender or environmental issues. However, and despite being already a considered dimension, it is so relatively new that there are still some objective questions that the ethical decision making cannot yet clearly answer.

The critical importance of ethical decision making in the context of AHA interoperable IoT ecosystems for wide communities [36], such as older adults, their carers, and the healthcare professionals, has been highlighted in the present work. The ultimate goal was to stress all the factors that might contribute to the consideration of ethical aspects in a large-scale pilot, which involved science, technology, and social issues such as big data, algorithms, the Internet of Things, older people’s activities, and caring solutions. Both theoretical and actionable mechanisms and practices had to be activated in the context of the highly demanding environment of the IoT vision to achieve successful outcomes of the study. A well-designed co-creative [37] assessment and evaluation process related to appropriate decision making, along with the ethical disclosure approach of science, strived to delve deeply into the ethical connotations.

To meet these ethical requirements, a detailed and elaborated ethical decision frame was defined, ensuring respect to human rights and the privacy of older adults.

Defining such a suitable ethical framework was not a simple or one-dimensional process, since IoT technologies, algorithms, big data, and interoperability ecosystems could not be considered neutral artefacts, and people had to decide about its use. Human competencies such as social responsibility and sensitivity to ethical issues along with practical methods needed to be explored to provide a decision support model to guide ethically aware choices and actions of researchers, developers, and clinicians.

As a main scientific output, this article indicates effective mechanisms orchestrated by an ethical framework that proved its robustness to cover diverse needs and contexts within a large-scale project. The aim of the present research work is to provide an ethically compliant decision support model by design [38] for researchers, developers, and clinicians that can be utilized within different IoT care environments.

However, some limitations of the present study challenged the ethical decision making of the pilot activities. More specifically, these limitations included: (a) data management and governance, and (b) the COVID-19 situation, which on one side brought concerns about communication means, whilst on the other side highlighted the added value of IoT interconnected services for the independency of older adults [39]. Additionally, the social distancing measures adopted during the COVID-19 pandemic required the employment of digital procedures for surveying participants, which imposed difficulties in user engagement that were reflected in the decision-making processes.

Research on ethical decision making in IoT large scale research is exploratory and the proposed model could be improved, taking into account similar strategies adopted by several projects which focus on healthy ageing solutions [40]. In addition, other enablers, apart from the participatory and deliberative process, could support future DSSs in terms of data management and participant rights preservation in large-scale research.

## Figures and Tables

**Figure 1 healthcare-10-00957-f001:**
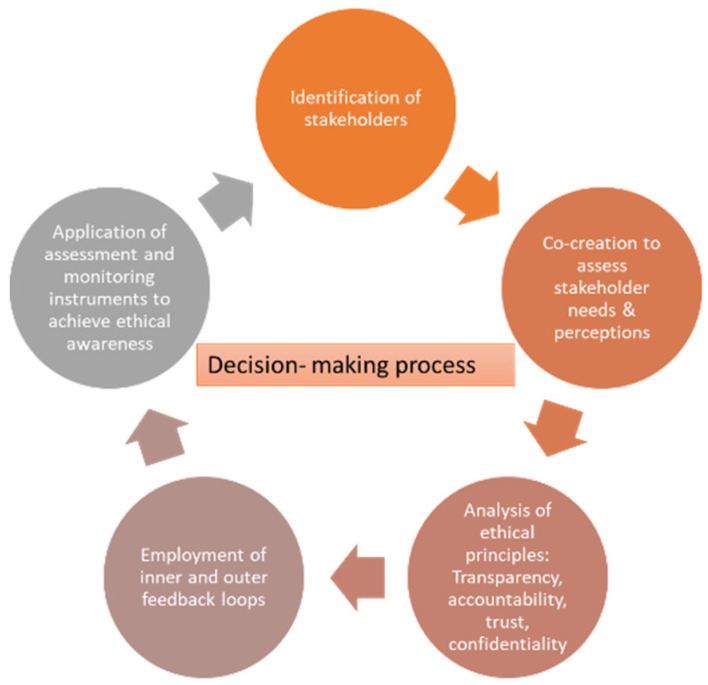
An Ethical compliant Decision-Making Framework for IoT.

**Figure 2 healthcare-10-00957-f002:**
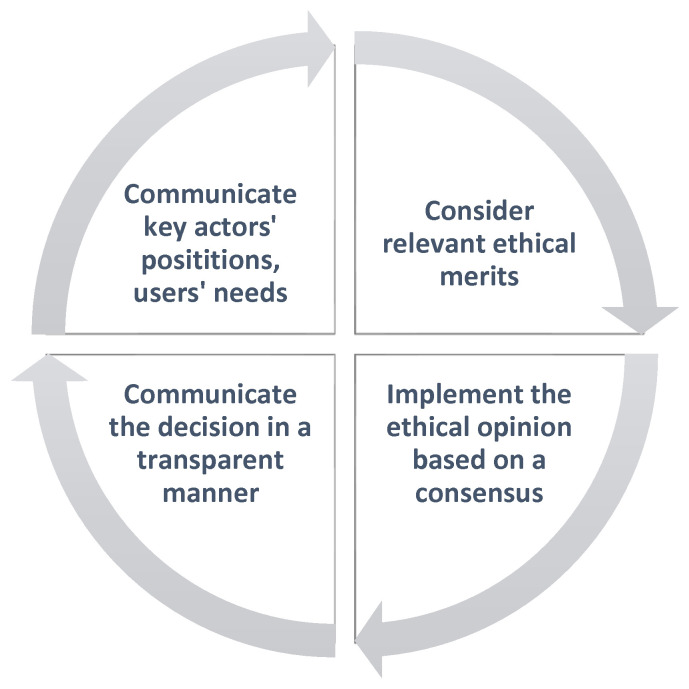
Inner and outer feedback loop mechanisms of ethical and sustainable decision-making process.

**Figure 3 healthcare-10-00957-f003:**
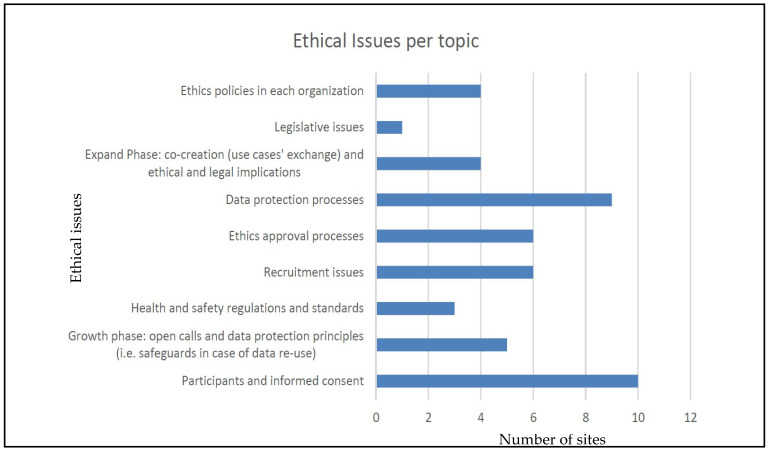
Number of declared issues per topic.

**Figure 4 healthcare-10-00957-f004:**
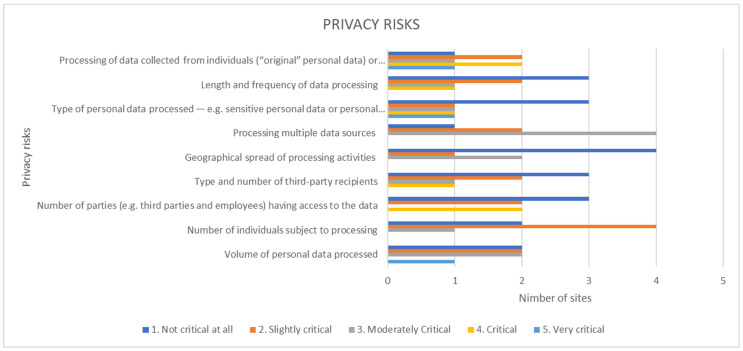
Data processing security-privacy risks.

**Figure 5 healthcare-10-00957-f005:**
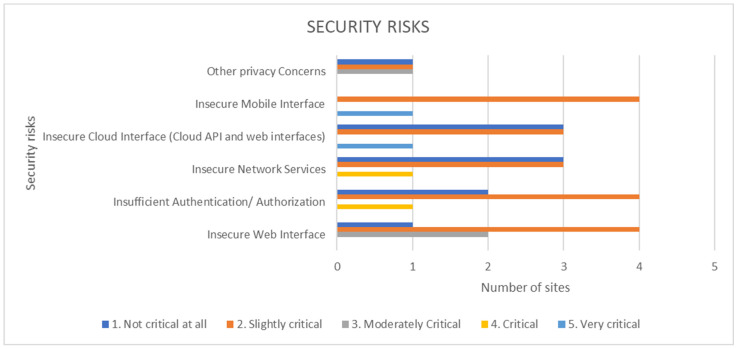
Data processing security risks.

**Figure 6 healthcare-10-00957-f006:**
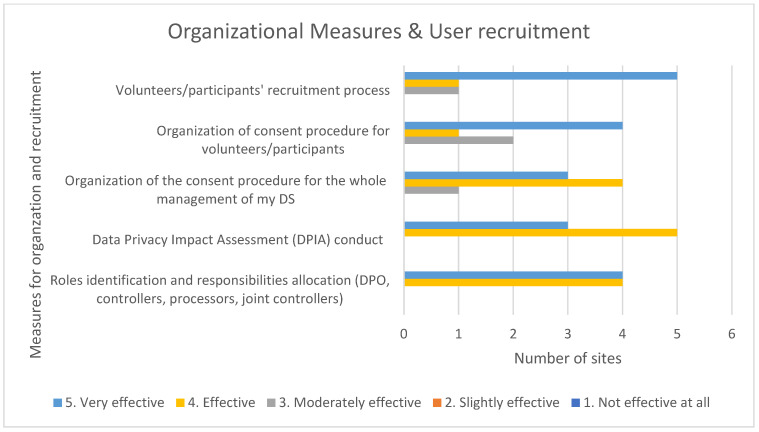
Organizational measures and user recruitment.

**Figure 7 healthcare-10-00957-f007:**
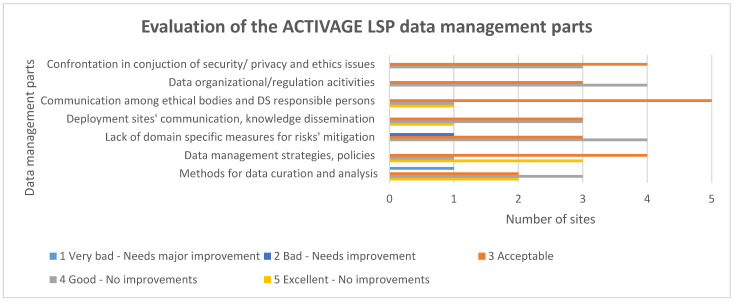
Evaluation of the ACTIVAGE LSP data management parts.

**Figure 8 healthcare-10-00957-f008:**
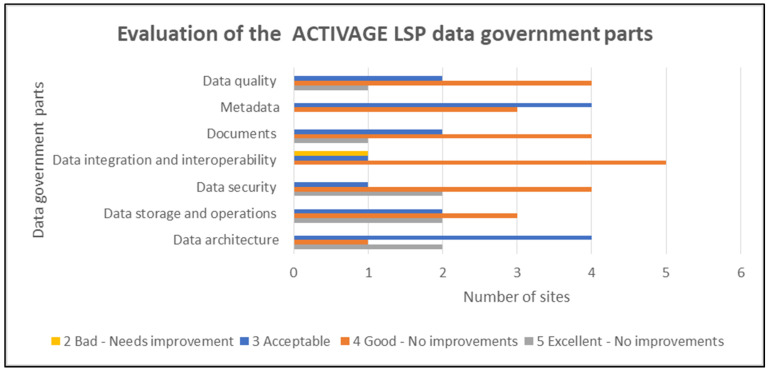
Evaluation of the ACTIVAGE LSP data government parts.

**Figure 9 healthcare-10-00957-f009:**
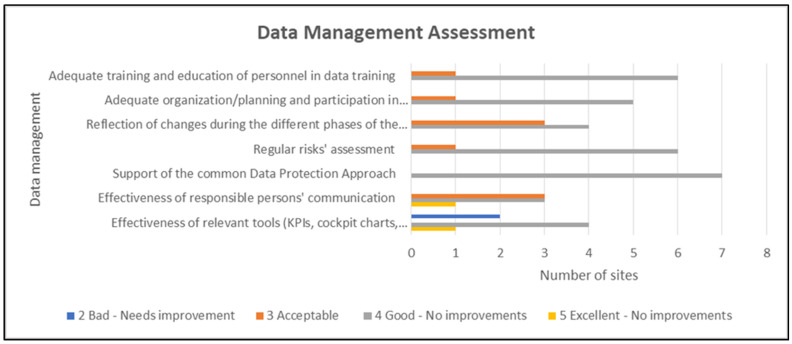
Data management assessment.

**Figure 10 healthcare-10-00957-f010:**
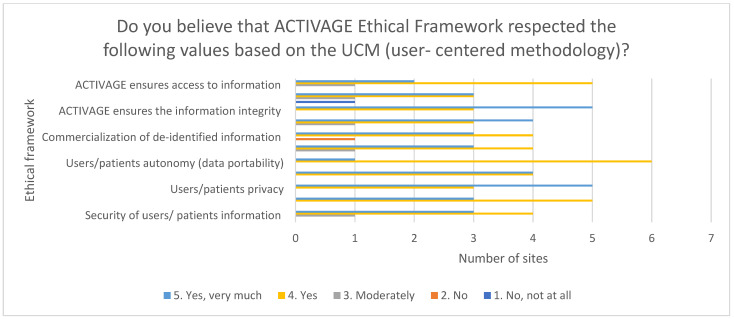
Final evaluation of ACTIVAGE Ethical Framework.

**Table 1 healthcare-10-00957-t001:** Deployment sites of the LSP.

Deployment Sites of the LSP
DS1 GAL: Galicia region (ES)
DS2 VLC: City of Valencia (ES)
DS3 MAD: Municipality of Madrid and Madrid Region (ES)
DS4 RER: Emilia-Romagna region (IT)
DS5 GRC: Digital Cities of Central Greece-CitiesNet (DCCG), Metamorfosi (MM), Pilea-Hortiatis (MPH), Mobility scenario (MM)
DS6 ISE: Isère region (FR)–Korian Institutions, Isère region (FR)–Homes
DS7 WOQ (DE): Weiterstadt, Treuchtlingen, Stuttgart, Rodgau, Türkheim
DS8 LEE: City of Leeds (UK)
DS9 FIN: Turku, Oulu, Tampere, Helsinki

## Data Availability

The data presented in this study are available on request from the corresponding author.

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
