# Peer review of "Ethical Decision Making in Iot Data Driven Research: A Case Study of a Large-Scale Pilot"

_healthcare, 2022, doi:10.3390/healthcare10050957_

Round 1

Reviewer 1 Report

Authors presented survey about ethical perspective in decision-making in IoT environment, focusing on risks and threats e.g. over-controlled environment. In article they addresses specific challenges of ‘smart living environments for ageing’

Strengths:
1) Clear presentation of Materials and Methods

2) Questionnaire for pilot sites included in article 

3) Author's the feedback loop mechanism of ethical decision-making process 

Improvements:

Page 3: "Error! Reference source not found."
Please provide source. 

Page 5
"external ethics and legal experts (xxx data analysts and xxx ethics experts)."
Please specify number of expert or proportion

Page 9 
"third parties’ recipients (Error! Reference source not found.).
Please provide source.

Page 12
"software developers often fail to address usability issues and user characteristics and preferences, ultimately implementing non  
user-friendly systems [25]."
Please provide more modern source then 2003 for such crucial issue

Page 16
"Defining such a suitable ethical framework was not a simple and one-dimensional process"
Please provide clear presentation of your practical ethical framework

Main issues:

1) Article goals are unclearly presented

2) What are the criteria for choosing pilot sites in project

3) What are the limitations of survey and study - please provide it

4) What contribution have authors in this article
"The proposed ethical framework was developed within the ACTIVAGE project and started with the analysis of the stakeholders needs" - There arise a question – have all the authors of this paper been a members of ACTIVAGE project from beginning? And the second question is have all the people who had significant impact on the final output been credited?

5) This article is ACTIVAGE project presentation. What is scientific output of this article?

6) Please include assessment of results from Figures 2-9, one sentence is not enough

Reviewer 2 Report

the article deals with very interesting topics, requires changes, but the idea is very good, the abstract does not reflect the content. suggests rebuilding them. The analysis of the literature is a good basis, but I recommend expanding it by several items taking into account other aspects of these solutions. The methodology is poorly described, it should be completely rebuilt the descriptions of the methods used, it is not enough. There is no information on how to select and try to make them justified. The results are described at a low level. There are no axis descriptions in any of the pictures. References to literature in the form of [error]. Bak to discuss and possible future research, research limitations.

Round 2

Reviewer 1 Report

I would like to thank the authors for taking the action to clarify concerns and to update the article to be more valuable.

Despite answering the below question kindly request the authors to include provided  answer within the article. The request is to clarify the participation and input of authors in this publication

“4) What contribution have authors in this article

"The proposed ethical framework was developed within the ACTIVAGE project and started with the analysis of the stakeholders needs" - There arise a question – have all the authors of this paper been a members of ACTIVAGE project from beginning? And the second question is have all the people who had significant impact on the final output been credited?”

Authors’ response: Thank you for your comment. It should be noted that each co-author has been involved in the context of their role in the project. More specifically co-authors have been involved with the 3 following roles a. pilot responsibles b. ethical managers c. scientific professional role: engineers, sociologists, health technicians, social workers etc. In respect to manuscript preparation, all authors were involved in the process of writing, methodology and editing. In addition, all the people who had significant impact on the final output have been credited in the paper.

Author Response

Authors’ response (2nd Round): Following the reviewer comment, at the current (updated) version of the manuscript, authors’ participation and input is explicitly stated at the end of the Conclusion Section (p.20) as quoted in the paragraph below:

“Conceptualization: Sofia Segkouli, Giuseppe Fico, Pilar Sila

Data Curation: Sofia Segkouli, Cecilia Vera-Muñoz, Pilar Sila, Matina Loukea

Formal Analysis: Sofia Segkouli, Mario Lecumberri, Mario Diaz Nava, Juan Bautista Montalvá Colomer

Methodology: Sofia Segkouli, Pilar Sila, Stefano Nunziata, Nadia Campanini, Enrico Montanari, Suzanne Morton, Maria Teresa Arredondo Waldmeyer

Validation: Mario Lecumberri, Antonis Voulgaridis, Andreas Triantafyllidis, Francesca Cocchi, Trinidad de Lorenzo, Eleni Chalkia, Sergio Guillén

Writing - original draft: Sofia Segkouli, Cecilia Vera-Muñoz

Writing - review & editing: Antonis Voulgaridis, Andreas Triantafyllidis, Alexander Dulcos, Matina Loukea

Supervision: Guissepe Fico, George Dafoulas, Konstantinos Votis

Project Administration: Guissepe Fico, Konstantinos Votis”

Reviewer 2 Report

The authors did not change much. Adding 1 literature item is a minor contribution. The authors' responses make sense, but should be included in the content of the authors, not in the response.

Author Response

The authors did not change much. Adding 1 literature item is a minor contribution. The authors' responses make sense, but should be included in the content of the authors, not in the response.

Authors’ response (2nd Round): Thank you for your comment, we have included all changes in the final version of the manuscript. More specifically in respect to Round 1 Review additional information have been added as follows:

- The abstract does not reflect the content, suggests rebuilding them: a. the previous version of the abstract b. the updated one.

Research that uses big data extensively through global interconnected infrastructures, such as ‘Internet of Things’ (IoT) for Active and Healthy Ageing (AHA), is fraught with several ethical concerns. This paper is the outcome of a collaborative multiscale research work, which had to address the complicated exercise of ethical decision making in IoT smart ecosystems for older adults. Diverse context and use cases in respect to chronic diseases required heightened attention for early ethical awareness in respect to the design and management of IoT experimental studies. IoT technologies generate intelligence, connectivity and develop knowledge to be used in the decision-making process. We claim that ethical decision-making in respect to the management of open and interoperable IoT systems and granular information of personal data has no clear-cut answers. Nonetheless, the present study illustrates how major IoT–related challenges were accomplished by the introduction and use of tangible mechanisms based upon a feedback loop mechanism in order to operationalize core ethical values such as transparency, trust and fairness in real care settings.

  1. IoT technologies generate intelligence, connectivity and develop knowledge to be used in the decision-making process. However, research that uses big data through global interconnected infrastructures, such as ‘Internet of Things’ (IoT) for Active and Healthy Ageing (AHA), is fraught with several ethical concerns. A Large-Scale application of IoT operating in diverse piloting contexts and case studies needs to be orchestrated by a robust framework, to guide ethical and sustainable decision-making in respect to data management of AHA and IoT based solutions. The main objective of the current article is to present the successful completion of a collaborative multiscale research work, which had to address the complicated exercise of ethical decision making in IoT smart ecosystems for older adults. Our results reveal that among the strong enablers of the proposed ethical decision support model were the participatory and deliberative procedure complemented by a set of regulatory and non-regulatory tools in order to operationalize core ethical values such as transparency, trust and fairness in real care settings for older adults and their caregivers.

Also keywords are modified accordingly

In the current version the abstract has been rebuilt in order to highlight better the main objective of the manuscript which is an ethical and sustainable decision –making in the LSP operation.

-The analysis of the literature is a good basis, but I recommend expanding it by several items taking into account other aspects of these solutions.

Thank you for your comment. In the current (updated) version of the manuscript, additional sources are added. More specifically, overall 11 new reference sources are included in the bibliography of the current version of the manuscript, marked with track changes the following sources are included in manuscript References

Also table 1 has been added to highlight the initial synthesis of the LSP before the open calls launch (p.3).

-The methodology is poorly described, it should be completely rebuilt the descriptions of the methods used, it is not enough.

Thank you for your comment.

  • Section 2 Materials and Methods, has been rebuilt by adding or modifying accordingly some paragraphs in order to stress core elements,

More specifically: changes in the 1st paragraph: The proposed ethical frame of decision making was developed on the basis of a bottom up approach, anchored in  the analysis of the a large number of stakeholders…… as well as European citizens comprised the large scale pilot.

Additional paragraphs: “The co-creation framework was introduced….IoT environment.

“Understanding and operationalizing …… awareness”.

“Outer feedback loops…. groups”.

“End-users….services’ use”.

“More specifically a concrete proportion….activities”

Modifications in figures’ caption: “Inner and outer feedback loop mechanisms of ethical and sustainable decision-making process”

Additing phrases to highlight the context of the decision making process: “that determined the decision–making process in a complex socio-technical context”.

Rewrite paragraphs to explain more clearly the design of the ethical decision making model:

“Nonetheless……evaluate”.

- There is no information on how to select and try to make them justified

Thank you for your comment. The main methods of the manuscript are the feedback loops that have already been applied in the LSP operations. The stages of the proposed decision making framework is mentioned in the following paragraph:

‘The proposed model of ethical decision making was composed of 4 stages: a. develop the frame b. assess through deliberative dialogue c. apply and update during the whole lifecycle of pilot activities and d. evaluate’. Furthermore the practically of these tools are demonstrated in the surveys that are presented in a detailed manner in the ‘Results’ section.

-The results are described at a low level.

Thank you for your comment. In order to address adequately this comment the section of Results has been rebuilt by removing paragraphs and adding new ones, for instance

Removing: “At the end….domain”

Adding: “A process of shared visions ….non-regulatory items. “After this assessment …practice”

“As it is evident from figure 2…behind it”.

-There are no axis descriptions in any of the pictures.

Thank you for your comment. Axis descriptions have been embedded in all pictures and also detailed explanation of the figures have been added in the present version: “According to figure 3… established for them.

“Regarding security risks… but there are technologies (i.e. block chains) to cover such risks”.

In addition, a subsection is added: 3.4 LSP concern in data management to highlight better the LSP data management aspects.

Additional paragraphs: “The same as the LSP data management… with no need of improvements”

“In general, the ACTIVAGE ethical framework was very well received..following the UCM”.

  -Back to discuss and possible future research, research limitations

Thank you for your comment. In order to highlight better future research and limitations, two paragraphs has been added: “However some limitations…. in turn in decision making processes”, Furthermore it has to be stressed that the research…..participants’ rights’ preservation in large scale research.
